# Bongkrekic Acid and *Burkholderia gladioli* pathovar *cocovenenans*: Formidable Foe and Ascending Threat to Food Safety

**DOI:** 10.3390/foods12213926

**Published:** 2023-10-26

**Authors:** Dong Han, Jian Chen, Wei Chen, Yanbo Wang

**Affiliations:** 1Beijing Advanced Innovation Center for Food Nutrition and Human Health, Beijing Engineering and Technology Research Center of Food Additives, School of Food and Health, Beijing Technology and Business University, Beijing 100048, China; donghan@btbu.edu.cn (D.H.);; 2Food Safety Key Laboratory of Zhejiang Province, School of Food Science and Biotechnology, Zhejiang Gongshang University, Hangzhou 310018, China

**Keywords:** *Burkholderia gladioli* pathovar *cocovenenans*, bongkrekic acid, food poisoning, foodborne pathogen, food safety

## Abstract

Bongkrekic acid (BKA) poisoning, induced by the contamination of *Burkholderia gladioli* pathovar *cocovenenans*, has a long-standing history of causing severe outbreaks of foodborne illness. In recent years, it has emerged as a lethal food safety concern, presenting significant challenges to public health. This review article highlights the recent incidents of BKA poisoning and current research discoveries on the pathogenicity of *B. gladioli* pv. *cocovenenans* and underlying biochemical mechanisms for BKA synthesis. Moreover, the characterization of *B. gladioli* pv. *cocovenenans* and the identification of the *bon* gene cluster provide a crucial foundation for developing targeted interventions to prevent BKA accumulation in food matrices. The prevalence of the *bon* gene cluster, which is the determining factor distinguishing *B. gladioli* pv. *cocovenenans* from non-pathogenic *B. gladioli* strains, has been identified in 15% of documented *B. gladioli* genomes worldwide. This finding suggests that BKA poisoning has the potential to evolve into a more prevalent threat. Although limited, previous research has proved that *B. gladioli* pv. *cocovenenans* is capable of producing BKA in diverse environments, emphasizing the possible food safety hazards associated with BKA poisoning. Also, advancements in detection methods of both BKA and *B. gladioli* pv. *cocovenenans* hold great promise for mitigating the impact of this foodborne disease. Future studies focusing on reducing the threat raised by this vicious foe is of paramount importance to public health.

## 1. Introduction

Foodborne diseases have been posing enduring societal and economic challenges worldwide. According to the World Health Organization (WHO), foodborne diseases are estimated to cause 600 million illnesses and 420,000 deaths every year, making them a major cause of morbidity and mortality globally [1,2]. These illnesses and deaths have a substantial impact on individuals, families, communities, broader society, as well as the global economy. In addition to the human distress caused by these diseases, they also have impactful influences on the food supply chain [2]. Therefore, effective prevention, control, and management of foodborne diseases remain a critical public health priority. Bongkrekic acid (BKA) is a potent respiratory toxin, which could significantly impair mitochondrial ATP/ADP exchange and has been implicated in numerous outbreaks of fatal food poisoning. It is classified as a flavorless, odorless, colorless, thermally stable, and highly unsaturated methoxy tricarboxylic acid [3]. Previous research has demonstrated that BKA is produced by *Burkholderia gladioli* pv. *cocovenenans* in natural conditions seeing that the BKA biosynthesis gene cluster (*bon*) exists in the gene repertoire of this bacterium [4,5,6]. Although the biosynthesis and pathogenesis of BKA are unique in many ways, the bacterium of *B. gladioli* pv. *cocovenenans* is prevalently distributed globally as it has been extensively isolated and identified in various food and environmental samples, such as water and soil, from all five most populated continents [7,8]. Results also indicated that many cultivated food ingredients, such as *Tremella fuciformis* (white wood ear mushroom), can be contaminated with *B. gladioli* pv. *cocovenenans* [9,10]. Since 1975, more than 3000 cases of BKA toxication have been officially documented with a remarkably high case-fatality rate (40–60%). Although a significant amount of previous BKA outbreaks occurred in Java, Indonesia, reports indicated that recent cases widely spread around in Africa, China, and Southeast Asia [3,6,11] (Table 1). A recent outbreak in Northeast China caused nine deaths, which has a case-fatality rate of 100% [12]. The toxicity of BKA relies on its occupation of a substrate binding site of mitochondrial ADP/ATP carrier to inhibit the activity of the carrier [13]. Considering the structural conservativity of mitochondrial ADP/ATP carrier in eukaryotic organisms, the toxicity of BKA is most likely universal across human and mammalian species. Previous research showed that BKA-contaminated diet can lead to the death of mice, dogs, and rhesus monkeys within 35 h [14,15]. Studies on oral administration have indicated LD50 value ranges of 1–3.16 mg/kg for human and 0.68–6.84 mg/kg for mice, respectively, while another study revealed that rats were able to survive an oral dose of 10 mg/kg of BKA, whereas a dose of 20 mg/kg was proved to be lethal [3,16].

Since the first report of a BKA outbreak in 1895, *B. gladioli* pv. *cocovenenans* contamination and BKA poisoning have been associated with a broad range of food products [23]. These food products include rice noodle [20], fermented beverages [7], fermented coconut (tempe bongkrèk) [24], rehydrated wood ear mushrooms (*Tremella* spp. and *Auricularia heimuer*) [9], fermented corn flour [12], fermented grains (glutinous rice, japonica rice, and corn), and sweet potato flour [3,23,25] (Table 1). It is understandable that the production and accumulation of BKA in *B. gladioli* pv. *cocovenenans* requires appropriate conditions for bacterial population proliferation, functional gene expression, and metabolite enzymatic biosynthesis [26,27]. The desired conditions for BKA production have been detected as near-neutral pH (6.5–8.0), mild temperature (22–30 °C), low NaCl concentration (lower than 2%), and suitable fatty acid composition (oleic acid rich), and these requirements can be accommodated by some fermented and rehydrated food matrices [9,14,28,29]. Any food-related microbiological studies are built upon pure-culture and co-culture experiments between microbes and food matrices. Nevertheless, most of such culture studies on *B. gladioli* pv. *cocovenenans* fermentation and BKA production were conducted decades ago [9,29,30].

The current detection method for BKA relies on high-performance liquid chromatography (HPLC)-based instrumental analysis, whereas real-time PCR techniques are primarily employed to detect the presence of the *bon* gene cluster, which is employed to verify the existence of *B. gladioli* pv. *cocovenenans* [4,31,32,33]. While these analyses have no problem generating scientific results and unveiling underlying mechanisms, it is important to consider their suitability for in situ and routine food production inspections, owing to the high cost and technical requirements associated with these assays. One conventional method of preventing BKA is to have a better manufacturing practice and quality control of these high-risk foods. However, to further enhance the effectiveness of these measures, additional efforts may be required [7]. Furthermore, emerging challenges such as climate change and the acceleration of antibiotic resistance pose significant threats to the food industry. These challenges can substantially increase the uncertainty of foodborne pathogens and diseases, making it even more crucial to implement effective preventive measures to safeguard consumer safety [34,35]. As a threat that is largely unnoticed by the general public while repeatedly proven to be lethal and difficult to detect, *B. gladioli* pv. *cocovenenans* contamination and BKA poisoning pose a legitimate hazard to consumers and could demolish their confidence in certain aspects of food consumption [36]. This review delves into the multifaceted research topic of BKA and *B. gladioli* pv. *cocovenenans*, aiming to provide valuable insights and a comprehensive blueprint to combat this persistent yet rising threat to public health.

## 2. A Dual Biography of *Burkholderia gladioli* pathovar *cocovenenans* and Bongkrekic Acid

At the beginning, *B. gladioli* pv. *cocovenenans* was referred to by the name of *Pseudomonas cocovenenans* [37]. In memory of American plant pathologist and microbiologist Walter H. Burkholder, the genus name of *Burkholderia* was assigned to seven bacterial species that were previously classified within the genus of *Pseudomonas* [38] (Figure 1). After then, this pathogenic bacterium was recognized as *Burkholderia cocovenenans*, according to genomic comparison study [39]. Later, phylogenetic analyses proved that *B. cocovenenans* is in fact a junior synonym to *B. gladioli*; therefore, its current nomenclature was conferred [40]. While the outbreaks of *B. gladioli* pv. *cocovenenans* frequently occurred in coconut-related food consumptions and particularly tempe bongkrèk, a traditional Indonesian fermented coconut food from which BKA derives its name, beforehand, the scientific understanding of this disease remained elusive until the 1930s [24,41]. Two scientists, van Veen and Mertens, carried out a series of studies on samples from central Java to explore the cause of the then-mysterious deadly food poisoning [42,43,44,45,46]. There were two poisonous substances discovered from *B. gladioli* pv. *Cocovenenans*-contaminated food matrices. Firstly, the researchers identified a yellowish bacterial pigment and named it toxoflavin(e) [44]. Toxoflavin is a type of pyrimidotriazine compound that can be produced by multiple *Burkholderia* spp. (Figure 2A). It has manifested antibiotic, fungicidal, and virulence enhancement functionalities and also exerts certain level toxicities toward eukaryotic cells [47,48,49]. Shortly after, bongkrek(ic) acid was purified and categorized. Although their molecular structures were investigated and proposed after their discovery, the structure of both toxoflavin and BKA was not conclusively determined until the late 1950s [37,50] (Figure 2B). In contrast to the relatively mild toxicity observed with toxoflavin, BKA exhibits a high level of toxicity in monkeys, with a fatal dose estimated to be approximately 0.5 mg per subject when administered orally to monkeys weighing 1–5 kg [46]. Despite evidence that proved that both toxoflavin and BKA were poisonous to human, BKA and its extreme respiratory toxicity have been broadly recognized as the singular cause for the life-threating symptoms given the fact that the toxicity of BKA overwhelmingly outweighs that of toxoflavin in nearly all measurable ways [3,11]. This inference gains further support from the fact that only *B. gladioli* pv. *cocovenenans*, along with BKA, has been associated with lethal outbreaks, whereas toxoflavin can be produced by numerous other bacterial strains that have not been linked to fatal food poisoning incidents [20,51]. It is worth noticing that the initial understanding of BKA and *B. gladioli* pv. *cocovenenans* was only derived from studies on Indonesian pathogenic strains, whereas the cause of outbreaks in China was verified to be the same afterwards [10].

## 3. Pathogenesis of Bongkrekic Acid

BKA has been verified to be extremely toxic toward every animal species investigated, such as monkeys, pigeons, rabbit, and rats, even when BKA is administered in non-pure form (fermented coconut cakes) [43]. The symptoms observed in animal BKA studies include initial hyperglycemia, subsequent hypoglycemia, and dramatic blood lactic acid content increases (2–3 times of the normal level), and these symptoms were later demonstrated to be associated with impaired mitochondrial oxidative phosphorylation [16]. Through respective experimental designs, researchers were able to conclusively determine that BKA could bind to adenine nucleotide translocator (ANT) which located on the mitochondrial inner membrane [24,52,53,54].

ANT, also known as ADP/ATP translocase or ADP/ATP carrier, is an important mitochondrial carrier that is responsible for transporting ADP into and ATP out of mitochondria [53,55]. ANT carriers out its role by cycling between two states: cytoplasmic open state (c-state) and matrix open state (m-state). At normal circumstances, free ADP from mitochondrial intermembrane space (from cell cytoplasm) can specifically bind to c-state ANT, whereas free ADP in mitochondrial matrix can bind to m-state ANT. These binds will lead to conversion cycles between c-state and m-state of ANT and allow the ADP/ATP exchange to be carried out (Figure 3A). When consumed by animals, owing to its apparent lipophilicity, BKA can firstly penetrate the inner mitochondrial membrane and then bind to m-state ANT to form a BKA-inhibited ANT structure (Figure 3B) [13]. The formation of this structure will prevent the eversion of m-state ANT into c-state and thus terminate respective ADP/ATP exchanges. In view of the efficient inhibitory activity of BKA (one BKA molecule bind to one ANT), robust stability of BKA-inhibited ANT, crucial role of ANT in eucaryotic metabolism, and conservativity of mitochondrial ANT, there are currently no effective medical or pharmaceutical treatments available to reverse the symptoms caused by BKA poisoning, although the invention and development of such treatments would undoubtedly be invaluable [3]. Therefore, current strategies for managing the risk of BKA and *B. gladioli* pv. *cocovenenans* outbreaks are primarily focused on preventive measures instead of medical treatments.

## 4. Genomic Characteristics of *Burkholderia gladioli* pathovar *cocovenenans*

### 4.1. Genomic Diversity and Prevalence

The previous generation of phylogenetic studies on pathogens relies on independent comparisons of live microorganisms or fixed genetic materials, such as cell protein gel electrophoresis, DNA–DNA binding, and detailed biochemical profiling, which was once the only viable method but was deemed as insufficient for both disease control and research purposes [40,56]. Not long ago, high-throughput sequencing and nucleic-tide-level genome documentation revolutionized this research field swiftly [57,58]. In silico genomic comparisons among online databases empowered researchers and medical professionals to conduct coherent and efficient genomic epidemiological analyses that were unimaginable previously [59]. Genomic assemblies confirmed that *B. gladioli* pv. *cocovenenans* incorporates two circular chromosomes with the *bon* gene cluster located at chromosome 1. It can be concluded from the genome of *B. gladioli* BSR3 and Co14 that the pathogenic strains also possess one clustered regularly interspaced short palindromic repeat (CRISPR) array sequence at chromosome 1 and a total of five copies of rRNA from the two total chromosomes (Figure 4). A group of researchers [8] conducted species-wide genomic comparative analysis on *B. gladioli* pv. *cocovenenans* with the focus on the *bon* gene cluster (Figure 4). This study offered a comprehensive perspective on global distribution and prevalence of *B. gladioli* pv. *cocovenenans*. The authors conducted analyses of genomic diversity and phylogenetic relationships using a dataset consisting of self-sequenced *B. gladioli* pv. *cocovenenans* Co14 genome and 238 published *B. gladioli* genomes available in the NCBI database. This study revealed that out of 239 *B. gladioli* genomes, 36 contained the *bon* gene cluster. The 15.06% occurrence of *bon* inserts it as a shell gene (15–95%) in pan-genome methodologies employed in this study, which conveys that the *bon* gene cluster with corelated BKA biosynthesis is not a species-wide bioactivity and acquirement of this gene cluster could be a recent event. The results also showcased average nucleotide identity (ANI) values between 97.29% and 100.00% among *B. gladioli* genomes, while these ANI values among genomes of *B. gladioli* pv. *cocovenenans* strains are in between 97.45% and 100.00%.

Alongside other *Burkholderia* spp. (both pathogenic and non-pathogenic), a pan-genome analysis was carried out on eight self-assembled genomes of *B. gladioli* pathovar *cocovenenans* by another recent research study [5]. The study demonstrated that *B. gladioli* pv. *Cocovenenans* exhibits an intricate population structure. Their results also indicated that the ancestor of the pathogenic *B. gladioli* gained the *bon* gene cluster (and respective pathogenesis) from horizontal gene transfer. Furthermore, a genome recombination event might cause the deletion of the *bon* gene cluster from the genome of *B. gladioli* pv. *cocovenenans*. This discovery suggested that the ancient *B. gladioli* may have obtained the *bon* gene cluster and related regulators from other species, and as they evolved, significant genetic divergence was observed among them. It can be referred from these results that the conventional ribosomal RNA-based identification method may not yield satisfactory discrimination efficacies for pathogen screening since limited core-gene differences exist between *B. gladioli* pv. *cocovenenans* and non-pathogenic *B. gladioli*. Accordingly, techniques targeting the *bon* gene cluster, such as PCR amplification and BKA production detection, could serve as practical identification methods.

### 4.2. The Bon Gene Cluster and Bongkrekic Acid Biosynthesis

Another study [4] employed the Lambda Red homologous recombination technique to establish a *bonA*-silenced mutant and confirmed that the biosynthesis of BKA is carried out by the *bon* polyketide synthase (PKS) gene cluster. Using bioinformatic tools, this study then comprehensively depicted the assemble processes and corresponding gene functions in BKA biosynthesis. The center of *bon* in strain *B. gladioli* pv. *cocovenenans* DSMZ 11318 comprises 3 open reading frames (ORFs), bonA, bonB, and bonD (equivalent to 4 ORFs—bonA, bonB, bonC, and bonD—in most other *B. gladioli* pv. *cocovenenans* strains) that encode modular type I PKS modules (Figure 3C,D). These PKS modules are essential to BKA biosynthesis since cycles of polyketide chain elongation take place at these modules. The *bon* gene cluster also comprises nine discrete gene loci, each encoding a free-standing protein to facilitate the biosynthesis of BKA. Specifically, since *bon* PKS lacks cognate acyltransferase, the loading of an extender unit to acyl carrier protein relies on free-standing cognate acyltransferase BonJ and BonK (Figure 5A) [60,61]. Similarly, enoyl reductase is absent from the modules in bonA-D, implying that enoyl reductions are carried out by another free-standing protein—BonE (Figure 5B). Researchers also noticed that BKA is rather uncommonly branched compared to many other bacterial PKS products; they deduced that β-branching occurs during the BKA assembly to introduce alkyl branches at the C-21 and C-3 positions in BKA. During the β-branching of BKA, BonF, BonG, and the duo of BonH and BonI most likely function as ketosynthase, 3-Hydroxy-3-methylglutaryl-CoA synthase, and enoyl-CoA hydratase, respectively (Figure 5C) [62]. After a typical PKS elongation process, the hydroxyl group at C-17 is methylated by BonM, an O-methyltransferase. Eventually, a novel cytochrome P450 monooxygenase (BonL) was inferred to be responsible for the introduction of a carboxyl group at C-22 (Figure 5D) [4].

## 5. Characteristics of *Burkholderia gladioli* pathovar *cocovenenans*

To date, studies on the microbial characteristics of *B. gladioli* pv. *cocovenenans*, including growth, metabolism, survival, stress resistance, gene expression, and BKA production, remain scarce. The lack of information can create challenges in preventive measures and interventions against this disease, and it can also result in unnecessary resource investments and increased risks to public health. Therefore, it is essential to increase current knowledge on its behavior and identify effective strategies to control its growth and spread. It is pivotal to elucidate the biosynthesis properties of the *bon* gene cluster and the metabolic characteristics of *B. gladioli* pv. *cocovenenans* in both defined culture conditions as well as in pertinent food matrices to profoundly address the rising risk caused by BKA and *B. gladioli* pv. *cocovenenans* [60,63].

### 5.1. Bongkrekic Acid Production in Culture Media

Researchers have conducted a series of investigations into the culture growth and BKA production characteristics of *B. gladioli* pv. *cocovenenans*, which have generated valuable insights into the various factors that influence the growth of this bacterium and production of BKA [29,64] (Table 2). Their studies have focused on identifying compounds with potential toxin-preventive capabilities to better understand how to control the spread of *B. gladioli* pv. *cocovenenans* and reduce the risk of BKA contamination. The study’s significant discovery is that the absence of either coconut oil or glycerol from the culture media completely halts BKA production, without affecting the growth of bacterial populations. This result highlighted the critical role of both coconut oil and glycerol in BKA synthesis and suggested that the fat composition and concentration present in the substrate significantly impacts BKA production. This finding was consistent with the results of another study, which demonstrated that the addition of oleic oil at a concentration of 3.31 mmol per gram of defatted rich coconut medium is a significantly favored substrate for BKA production [9]. The observation indicated that oleic oil, a type of monounsaturated fat that is commonly found in olive oil, might be readily metabolized by *B. gladioli* pv. *cocovenenans* for BKA synthesis, resulting in a high level of BKA production. This suggests that specific lipid formation can significantly impact BKA production, with oleic oil being particularly effective. The significant influence of coconut oil and fat content, likely leading to increased Bongkrekic acid (BKA) production, may explain earlier BKA poisoning cases in coconut-related foods. Conversely, more recent BKA outbreaks, predominantly linked to corn- and rice-based foods with lower fat concentration, may be attributed to factors such as pathogen transmission across regions, prolonged fermentation periods, and varying nutrient compositions in these respective foods. This observation underscores the importance of unraveling the complexities of BKA-related incidents in diverse food contexts. The results of previous studies [29,64] also demonstrated that supplementing culture medium with either 2% NaCl or acetic acid (adjusted to pH 4.5) was effective in reducing the formation of BKA (Table 2). However, neither of these supplementations was able to completely inhibit the BKA production alone. Interestingly, the combination of 2% NaCl and acetic acid (adjusted to pH 4.5) was found to have a sufficient inhibitory effect on BKA production by inhibiting the bacterial growth, reducing the concentration of BKA to below the limit of detection (10 μg/g) using HPLC. This observation was consistent across all three investigated *B. gladioli* pv. *cocovenenans* strains, which initially had bacterial populations ranging from 5.78 log CFU/mL to 7.04 log CFU/mL. The findings of this study suggest that a combination of NaCl and acetic acid has a robust inhibitory effect on BKA production, and respective hurdle strategies could be viable options for reducing the risk of BKA contamination.

### 5.2. Effect of Food Ingredients on the Production of Bongkrekic Acid

In the same study [29,64], the authors also evaluated the anti-BKA production activities of four natural spices: garlic powder, onion power, capsicum power, and turmeric power. The study found that adding 0.6% garlic powder, 0.6% onion powder, 0.8% capsicum powder, and 0.6% turmeric powder to coconut culture medium could inhibit the formation of BKA when the initial populations of *B. gladioli* pv. *cocovenenans* were low (between 3.64 and 5.27 log CFU/mL). However, when the initial populations of *B. gladioli* pv. *cocovenenans* were high (above 7.32 log CFU/mL), adding up to 2% of any these spice supplements did not completely inhibit the BKA production. Additionally, the study revealed that the most effective way to utilize these spices is when the population of *B. gladioli* pv. *cocovenenans* is at minimum level. The results suggested that natural spices can be effectively used in applicable food products to inhibit the growth of *B. gladioli* pv. *cocovenenans* and prevent BKA accumulation when utilized properly. The study’s findings are of great importance as they have the potential to aid in disease prevention efforts in both food manufacturing and consumption. Although there has been more recent experimental research on *B. gladioli* pv. *cocovenenans* and BKA that heavily focused on toxin and microbe detections, the significance of culture studies on food matrices lies in their unique approach to the problem. The scarcity of similar studies in the current research landscape highlights the need for additional research in this area. Such research can help in developing novel and effective strategies for BKA poisoning prevention.

### 5.3. Bongkrekic Acid Production under Co-Culture Conditions

Beneficial and non-pathogenic microorganisms play a vital role in preventing the development of foodborne pathogens and safeguarding food safety [65]. For instance, by occupying the same ecological niches with harmful microorganisms, lactic acid bacteria, such as *Lactobacillus* spp. and *Pediococcus* spp., can compete for nutrients and produce antimicrobial substances that inhibit the growth and activity of foodborne pathogens [66]. Although present naturally in a diverse range of food commodities, these microbes can also be deliberately introduced or promoted under preferable culture conditions to achieve their dominance in respective food matrices. Moreover, the growth kinetics and antagonistic effects between these microbes and foodborne pathogens, e.g., *Salmonella* spp., *Campylobacter jejuni*, *Yersinia enterocolitica*, pathogenic *Escherichia coli*, *Shigella* spp., *Vibrio* spp., and others, have been previously depicted by employing co-culture experiments [67]. However, co-culture studies regarding *B. gladioli* pv. *cocovenenans* and BKA production have been rare, considering that the overwhelming majority of BKA outbreaks that occurred in food matrices consist of complex microbial communities. In addition to examining other common factors, the previous study also investigated the impact of *Rhizopus oligosporus*, the fungus traditionally used for fermenting tempe bongkrèk, on the growth of *B. gladioli* pv. *cocovenenans* and the production of BKA under co-culture conditions [29]. In agreement with earlier research, the results demonstrated that although *R. oligosporus* possesses certain inhibition effects against the growth of *B. gladioli* pv. *cocovenenans*, the BKA synthesis reduction activities require a substantially higher initial population of *R. oligosporus* compared with that of *B. gladioli* pv. *cocovenenans* [14,68]. One intriguing finding in these studies was that the presence of organic acid with a pH of 4.5 was one of the most effective inhibitors of BKA production, resulting in complete inhibition under almost all culture conditions. It is noteworthy that during the fermentation of high-protein foods, *R. oligosporus* may result in pH increases instead of decreases. For instance, a particular study demonstrated a significant rise in pH value from 6.30 to 7.18 in soybean fermented with *R. oligosporus* after 72 h of fermentation [64,69]. This observation may offer a partial explanation for the frequent outbreaks of BKA in tempe bongkrèk.

Despite undergoing clear acidification during fermentation, standing in contrast to tempe bongkrèk’s near-neutral pH values, many acidic foods have been linked to BKA outbreaks [7,12,70]. Although the exact cause of this phenomenon is not completely understood, we hypothesize that it could be caused by differences in microbial characteristics involved in the fermentation process, variations in initial microbial load, fluctuations of fermentation conditions, or other unknown factors. Future research efforts could focus on characterizing the microbial composition and dynamics of different fermentative environments, investigating the impact of fermentation conditions (e.g., temperature, pH, salt concentration, and water activity), and identifying potential techniques that may reduce or inhibit BKA production. Additionally, these studies offer insights for developing intervention strategies, such as adjusting starter cultures or standardizing fermentation conditions, to reduce the risk of BKA contamination in food.

## 6. Detection and Analytical Advancements

The dependability of detection methods plays a pivotal role in establishing the credibility of any epidemiologic or academic research. In the context of foodborne illnesses like BKA and *B. gladioli* pv. *cocovenenans*, binary detection, microbial analysis on the pathogen or chemical analysis on BKA, appears to be an appropriate and viable approach, given the nature of the research subject.

### 6.1. Detections of Bongkrekic Acid

From the very beginning, the identification and validation of BKA was conducted using paper chromatography methods, which can accurately detect a relatively pure form of BKA at levels as low as 0.05 μg [37,50,71]. Later, with technological advancements, more advanced analytical methods such as HPLC and mass spectrometry (MS) have become available (Table 3). These instrumental techniques offer enhanced accuracy and sensitivity when compared to paper chromatography, allowing for more efficient and dependable analysis of BKA in both pure formation and in food matrices [33,72]. The use of the respective methods has facilitated the detection of trace amounts of BKA in complex food matrices, enabling the conduction of highly sophisticated scientific research and reliable epidemiological analyses. With the help of these studies, researchers have acquired a better understanding of BKA in food products [3]. Despite their advantages, it is worth noting that equipment installation and maintenance for these analyses can be costly, requiring specialized technical expertise and corresponding sample preparation procedures. These characteristics make them time-consuming and challenging to conduct in situ application and routine food safety testing.

To address the limitations posed by instrumental methods, researchers have turned to alternative techniques that offer greater simplicity, are user-friendly, and are more compatible for in situ food analysis. One major category of these methods is immunoassays, which can quantify BKA based on developed antibodies in theory. The respective immunoassays, such as colloidal gold-based immunochromato-graphic assay (GICA) and enzyme-linked immunosorbent assay (ELISA), have been developed and validated by few Chinese institutions for rapid BKA detection in food products and other types of samples [6,73]. These methods rely on the robustness of the antibody–antigen reaction to enable efficient and reliable detection of BKA, thus providing a valuable tool for food safety monitoring and quality control. Future studies can explore the applicability of these methods in a broader range of samples and in different settings. Additionally, studies can be conducted to evaluate the feasibility of integrating these methods into existing food safety monitoring systems, as well as the cost-effectiveness of implementing them on an expanded scale.

### 6.2. Detections of Burkholderia gladioli pathovar cocovenenans and Gene

Culture-based analytical methods are crucial for ensuring the accuracy and reliability of the microbial research findings since the culturability of microbes is important to both scientific discovery and prolonged laboratory microbial preservation. Therefore, conventional microbiological analytical approaches, including non-selective medium enrichment, subculturing on differential media, colony observation, microscopic morphology, biochemical tests, gram staining, and toxicity evaluation, have been broadly adopted by research on *B. gladioli* pv. *cocovenenans* [6]. The respective analytical protocols were both documented by scientific studies and standardized by government authorities, such as the National Standard of the People’s Republic of China (GB 4789.29–2020) [6,74]. It Is a justifiable inference that these protocols are labor-intensive, and improvements in these assays could result in substantial benefits in respective studies and regulatory tests.

Considering the challenges involved in using traditional culture-based assays, exploring alternative approaches for detecting the presence of *B. gladioli* pv. *cocovenenans* could be advantageous. The utilization of novel nucleic-acid-based techniques presents a promising alternative, as they are capable of providing enhanced accuracy and sensitivity in comparison with conventional methods. Therefore, these techniques could be considered as a viable option to identify the presence of *B. gladioli* pv. *cocovenenans* [75,76]. Previous studies have primarily focused on the use of qPCR or novel recombinase polymerase amplification techniques to detect *B. gladioli* pv. *cocovenenans* targeting the 16S–23S rRNA-encoding regions [74,77,78]. These methods have shown a high sensitivity in detecting *B. gladioli* pv. *cocovenenans* with a detection limit as low as 100 CFU/mL, which can be attributed to the advancements in molecular biology technologies [77]. It should be noticed that the identification efficacy validations of these assays were carried out in between *B. gladioli* pv. *cocovenenans* and other genera of bacteria, rather than non-pathogenic *B. gladioli* strains and other *Burkholderia* spp. strains. In addition to its current application for distinguishing between bacterial species, 16S–23S rRNA genes can potentially serve as a valuable tool for species-level identification [79,80]. Future studies could be conducted to explore the utilization of 16S–23S rRNA genes for more precise species-level identification, which holds promise for enhancing the accuracy and comprehensiveness of *B. gladioli* pv. *cocovenenans* and other bacterial identifications. Given the presence of the BKA synthetic *bon* gene cluster and its 15.06% occurrence in recorded *B. gladioli* genomic assemblies [8], the *bon* gene cluster stands as one of the limited distinguishable genetic disparities between *B. gladioli* pv. *cocovenenans* and other *B. gladioli* strains, and we propose that nucleic acid-based tests targeting specific fragments of *bon* could offer highly reliable and accurate approaches for identifying *B. gladioli* pv. *cocovenenans*. Such tests have the potential to be highly specific and sensitive in the detection and diagnosis of this pathogen.foods-12-03926-t003_Table 3Table 3Detection method for BKA or *B. gladioli* pv. *cocovenenans* in foods.Food MatrixTargetAnalytical MethodLimit of DetectionReferenceGlutinous rice flour, corn flour, tremella (white wood ear mushrooms)BKAMixed-mode weak anion exchange solid-phase extraction combined with HPLC and diode array detection30 μg/kg[81]Formula rice powderLiquid chromatography–electrospray ionization quadrupole time of flight mass spectrometry2 μg/kg[82]Liquor fermentation cultureUPLC–MS/MS0.4 μg/kg[83]Rice noodleHPLC-Orbitrap High-Resolution MS with a Fe3O4/Halloysite nanotubes procedure0.3 μg/kg[33]*Auricularia heimuer* (black wood ear mushroom)HPLC–MS/MS with novel sample preparation technique1.0 μg/kg[84]Rice, corn flour, and fermented corn noodleUPLC-MS/MS0.12 μg/kg[72]tremellaVisual detection using cysteamine modified gold nanoparticles3.43 nM[85]*Auricularia heimuer*, tremella, and rice noodleColloidal gold immunochromatography assay1.2 μg/k[73]tremella*B. gladioli* pv. *cocovenenans*Loop-mediated isothermal amplification technology on 16S–23S rRNA-encoding region76 CFU/mL[78]Glutinous rice soupQuantitative PCR on 16S–23S rRNA-encoding region361 CFU/mL[74]Rice noodle, fresh white noodle, and glutinous rice flour Pathogenic and non-pathogenic *B. gladioli*Recombinant enzyme polymerase amplification with CRISPR/Cas12a system10–100 CFU/mL[77]HPLC: High-performance liquid chromatography; UPLC: Ultra-performance liquid chromatography; MS: mass spectrometer; CRISPR: clustered regularly interspaced short palindromic repeats.

## 7. Conclusions

In summary, the contamination of *B. gladioli* pv. *cocovenenans*, leading to the development of bongkrekic acid (BKA) poisoning, has emerged as a critical and lethal food safety concern with dire consequences, posing significant challenges to public health. The incidence of BKA poisoning in recent years has ignited a pursuit for a more profound comprehension of the pathogenicity of *B. gladioli* pv. *cocovenenans* and the underlying biochemical mechanisms for BKA synthesis. Specifically, characterization of *B. gladioli* pv. *cocovenenans* and the identification of the *bon* gene cluster have provided an essential foundation for developing targeted interventions to prevent and control BKA accumulation, while high-throughput sequencing and in silico genomic comparisons have unveiled the origin and global prevalence of this pathogen, highlighting this foodborne disease as a significant and ascending threat. Although limited, previous research based on culturing techniques has provided evidence that *B. gladioli* pv. *cocovenenans* is capable of producing BKA in different environments, which highlights the possible food safety hazards associated with BKA poisoning. Due to the scarcity of information regarding the impact of culture conditions such as pH, salt content, antimicrobial agents, and coexisting microorganisms on BKA reduction, it is crucial to enhance our understanding of inhibiting BKA production and promote respective applications to ensure food safety. Advancements in the detection methods of both BKA and *B. gladioli* pv. *cocovenenans* hold promise for mitigating the impact of this foodborne disease. Overall, it is imperative to undertake future research initiatives on BKA and *B. gladioli* pv. *cocovenenans*. Such research endeavors hold significant potential in controlling the threat posed by this formidable adversary to public health.

## Figures and Tables

**Figure 1 foods-12-03926-f001:**
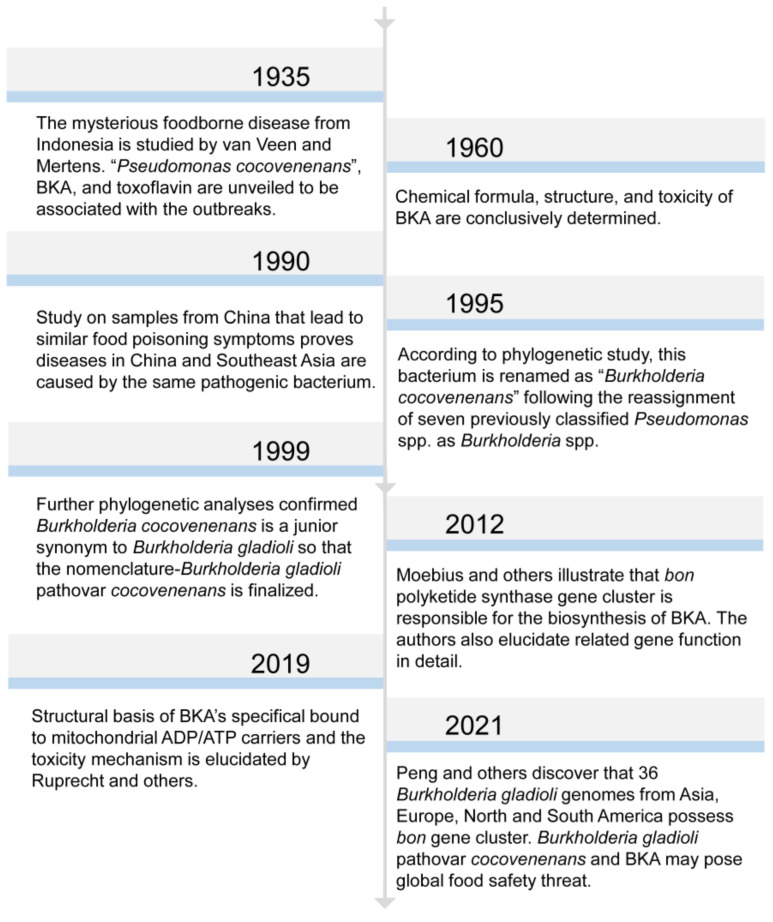
Chronological list of significant research findings in BKA and *B. gladioli* pv. *cocovenenans*. The presented years are literature publication years.

**Figure 2 foods-12-03926-f002:**
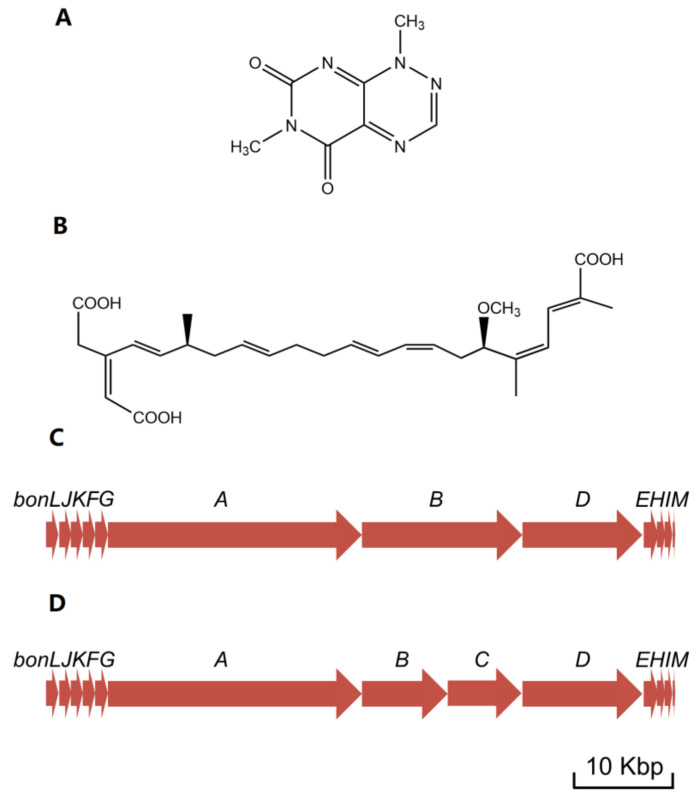
Illustration of related molecular structures and gene clusters. (**A**) Structure of toxoflavin; (**B**) Structure of bongkrekic acid; (**C**) Bongkrekic acid synthesis gene cluster in *B. gladioli* pv. *cocovenenans* DSMZ 11318; (**D**) Bongkrekic acid synthesis gene cluster in *B. gladioli* pv. *cocovenenans* BSR3. The presented length of a gene is directly proportional to the quantity of its base pairs.

**Figure 3 foods-12-03926-f003:**
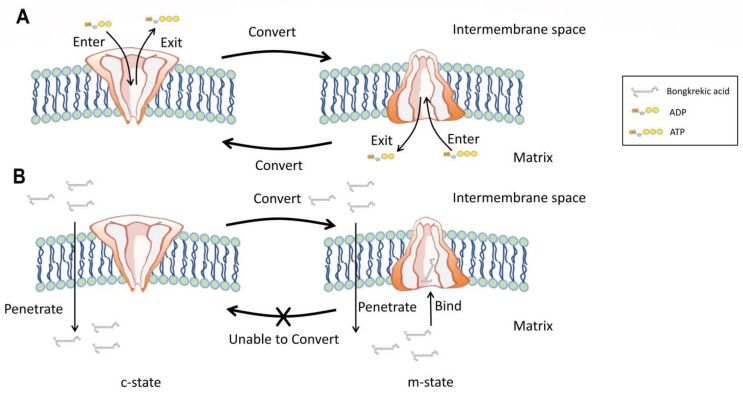
Pathogenic mechanism of BKA. (**A**) Unimpaired physiological function of mitochondrial ADP/ATP carrier. (**B**) Function of mitochondrial ADP/ATP carrier impaired by BKA.

**Figure 4 foods-12-03926-f004:**
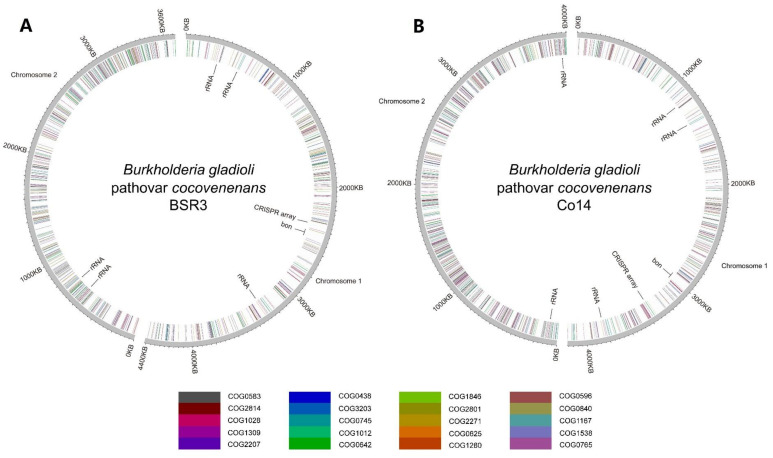
Circular genome diagrams of (**A**) *B. gladioli* pv. *cocovenenans* BSR3 and (**B**) *B. gladioli* pv. *cocovenenans* Co14. Genes of the top 20 abundant clusters of orthologous groups (COGs) are color-labeled according to their COG categories at the respective gene loci. The rRNA gene loci, CRISPR array locus, and *bon* gene locus are labeled at the inner circle.

**Figure 5 foods-12-03926-f005:**
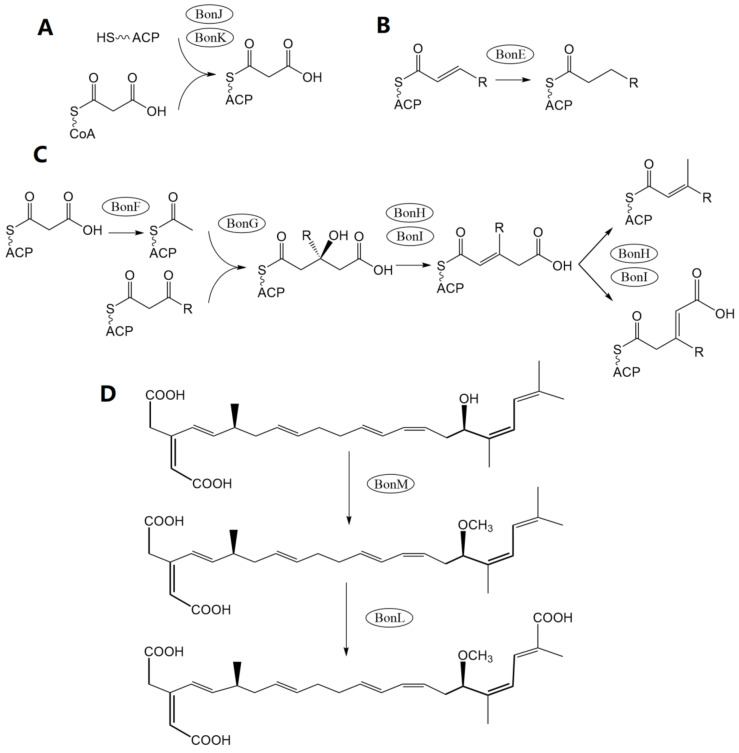
Depiction of essential free-standing gene functions in the biosynthesis of BKA. (**A**) The loading of the chain-extending unit to acyl carrier protein via cognate acyltransferase BonJ and BonK. (**B**) Enoyl reductions carried out by enoyl reductase BonE. (**C**) β-branching carried out by ketosynthase BonF, 3-Hydroxy-3-methylglutaryl-CoA synthase BonG, and enoyl-CoA hydratase BonH and BonI. (**D**) The terminal synthetic reactions are executed by the catalytic activity of O-methyltransferase BonM and P450 monooxygenase BonL.

**Table 1 foods-12-03926-t001:** Documented bongkrekic acid poisoning cases.

Region (Country)	Year ^a^	Food Matrix	Case-Fatality Rate (Deaths vs. Number of Illnesses)	Reference
Central Java (Indonesia)	2007	Fermented soybean pulp	33.3% (10:30)	[17]
Yunnan Province (Southern China)	2014	Fermented corn flour snacks	22.7% (5:22)	[18]
Southern Africa (Mozambique)	2015	Brewed corn flour alcoholic beverage	32% (75:234)	[19]
Guangdong Province (Southern China)	2018	Rice noodle (not fermented or spoiled)	50% (2:4)	[20]
Guangdong Province (Southern China)	2019 (Publication year)	Rice noodle (expired)	60% (3:5) (5 separated cases)	[21]
Heilongjiang Province (Northern China)	2020	Fermented corn flour	100% (9:9)	[12]
Dagana District (Bhutan)	2020	Brewed corn alcohol (suspected)	66.7% (4:6)	[22]

^a^ Refer to the year in which the outbreak occurred unless stated otherwise.

**Table 2 foods-12-03926-t002:** Substrate concentrations that resulted in substantial BKA production in culture media.

Substrate Concentration ^a^	Medium Type ^b^	Culture Medium Formulation	Initial Bacterial Population (Log CFU/g or Log CFU/mL)	Bacterial Growth When the Substrate Exceeds the Concentration Range ^c^	Reference
Between 0 and 10% glycerol	Wet	Coconut culture medium: prepared from water-pressed desiccated coconut with a final pH 6.9	5.96	+	[64]
Lower than 4% glucose			4.54	+	
Lower than 0.8% garlic powder			4.64	Unknown	
Lower than 0.6% onion power			5.27	Unknown	
Lower than 0.8% capsicum power			3.64	Unknown	
Lower than 0.6% turmeric power			5.27	Unknown	
Lower than 2% NaCl and pH value higher than 4.5 (adjusted using acetic acid)			5.78 to 7.04	−	[29]
Higher than 10% coconut fat	Dry	Defatted rich coconut medium: prepared from water-blended fresh coconut meat with lipid content extracted	8.95	+	[9]
3.31 mmol/g oleic acid	Wet			+	
Higher than 40% lauric acid	Dry			+	

^a^: The concentration of a certain substrate at which substantial bongkrekic acid production was detected; ^b^: The type of medium (wet or dry) used for calculating the substrate concentration; ^c^ +: apparent bacterial growth was observed, −: no apparent bacterial growth, Unknown: the bacterial growth was not measured or mentioned in the culture media.

## Data Availability

The data used to support the findings of this study can be made available by the corresponding author upon request.

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
