# Peer review of "Bongkrekic Acid and Burkholderia gladioli pathovar cocovenenans: Formidable Foe and Ascending Threat to Food Safety"

_foods, 2023, doi:10.3390/foods12213926_

Round 1

Reviewer 1 Report

Comments and Suggestions for Authors

I read this review with pleasure and I congratulate the authors. The topic has not been much explored outside of Asia because the bacterium actually does not cause many problems with foods in other continents. In any case, it is important to stay up to date on all food safety topics. The review has been set up correctly and provides all the useful elements for those who have to deal with this bacterium and the human health problems it can cause. IN THE FILE I ATTACH (I ask the authors to see it) I have reported small typos and suggestions for additions to the text.

Comments on the Quality of English Language

The article is written in good, readable English. IN THE FILE I ATTACH I have reported small typos and suggestions for integration.

Author Response

Response: Dear Reviewer, we appreciate your kind and valuable comments regarding our manuscript. Our review indeed aims to raise more awareness about the potential threats posed by Bongkrekic acid (BKA) and Burkholderia gladioli pathovar cocovenenans. By doing so, we hope to contribute to a more comprehensive understanding of food safety issues. Your suggestion is well-received, and significant effort has been made to revise our manuscript according to the comments. We would like to address your comments individually as following:

“I assume that the authors are referring to Saccharina japonica which is actually widely consumed in Asia. Perhaps it is useful to specify better what it is, for those who don't know.”

Response: thank you for the insightful observation. Here, we aim to highlight japonica rice (Oryza sativa subsp. japonica), a commonly consumed rice species in Asian countries. Japonica rice, along with glutinous rice (Oryza sativa subsp. glutinosa) and corn has been found to be associated with BKA poisoning in fermented grains [1]. In response to the reviewer's comments, we have revised the phrase as: "fermented grains (glutinous rice, japonica rice, and corn)" This revised sentence would avoid any potential confusion regarding these rice varieties (Line 75-76).

“In italic, please.”

Response: we sincerely apologize for the errors regarding the italicization of binomial nomenclature throughout our manuscript. Such errors are truly unacceptable in academic literatures. We assure that we have carefully checked and properly italicized the nomenclature at here and all other places. Thank you very much for your assistance and guidance in improving the quality of our manuscript. We are committed to upholding the highest standards of academic writing in our work.

“I think that in the individual boxes the verbs are all put in the present tense. In my opinion the overall meaning of the Table is better conveyed.”

Response: thank you. We have converted all the verbs into the present tense. Additionally, we have updated the overall artistic style to enhance the visual presentation of the table (Line 142-144).

“In the parts highlighted here in yellow, the letters "a" and "b" must be placed in superscript.”

Response: "a" and "b" have been correctly formatted into superscript style. The input is greatly appreciated, and it has helped us ensure the accuracy of our document (Line 338-339).

“I imagine that the letters "a" and "b" should be placed in superscript, but what reference is this referring to?”

Response: The use of "a" in that table was indeed a mistake and has been removed. Much thank (Line 455-456).

“Perhaps it is useful to specify better what Auricularia is”

Response: thank you so much for the kind suggestion. We have thoroughly revised the description concerning tremella (which includes various types of white wood ear mushrooms within the Tremella genus) and Auricularia heimuer (a specific species of black wood ear mushroom) in Table 3 (Line 455-456).

References:

  1. Zhang, H.; Guo, Y.; Chen, L.; Liu, Z.; Liang, J.; Shi, M.; Gao, F.; Song, Y.; Chen, J.; Fu, P. Epidemiology of foodborne bongkrekic acid poisoning outbreaks in China, 2010 to 2020. PLoS One 2023, 18, e0279957.

Reviewer 2 Report

Comments and Suggestions for Authors

The structure of the writing is difficult to comprehend (sentences are usually long and complex).  The review should give sufficient background to the reader without the need to extensively go to the cited reference for information. Scientific facts should be carefully extracted from the references and carefully presented. After reading some of the references, I see a significant association between B. gladioli (and its toxin) and Rhizopus spp in fermented food. One reference also provided very good information on the taxonomy of the bacterium and its pathogenicity. The review should briefly introduce the bacterium such as how many species are in this genus, how many serovars are in this species, how the serovars are differentiated, are they all considered food safety concerns.  A reference pointed out that the genes responsible for BA production, bon, were recently acquired and presented in only 15% of the analyzed B. gladioli. This fact should be elaborated on since it can affect in assessment of food safety. Characteristics of foods in which the BA can accumulate to the fatal level should be discussed in more detail and practical preventive measures should be suggested specifically. Although BA poisoning is devastating, it occurs sporadically and is associated with only some food types, i.e., traditional foods in poorly sanitary settings, control measures should be suggested accordingly. It was stated on several occasions that this is a rising threat and of global concern, the authors should provide more convincing evidence supporting this statement.

Some specific comments:

L.43: The word “respiratory toxin” should be explained in more detail

L.44-45: references should be cited for this statement

L. 45-47: the reference cited did not support this sentence (bon “only existed in this bacterium”) to claim this statement, other bacteria should be assessed for this gene and found to be negative.. maybe from other reference, please check

Table 1: what is “noticeable” means in this context and why the main food, i.e., fermented coconut pulp was not in the list?

Figure 1: this figure should not be listed as a “framework”, it’s a list of findings. What is “certified literature” means in this context?

Figure 3: Need more detail explanation. All arrows should be explained.

L. 204: This reference did not state this sentence.

Figure 4: Please check for accuracy. The reference stated that no bon cluster in chromosome I in BSR3

L.246-247: Please check for accuracy

L. 286-288: stated that coconut oil and glycerol significantly impact the BKA production, however, Table 1 shows the occurrence of outbreaks caused by non-fatty food such as corn flour, rice noodles, and alcoholic drinks. This conflict should be discussed.

Table 2 and its explanation in the main text is difficult to understand.

Comments on the Quality of English Language

There are many long sentences that are difficult to read and understand. Non-specific words such as this, that, those, etc. should be used at a minimum since it sometimes causes confusion of what that actually means (especially in the long and complex sentence). The document should be checked for typos and spelling throughout. The use of redundant wording should be limited. Concise expression is preferred. 

Author Response

Response: Dear Reviewer, we sincerely appreciate your valuable comments regarding our manuscript. Our primary objective is to enhance awareness of the potential risks associated with Bongkrekic acid (BKA) and Burkholderia gladioli pathovar cocovenenans. We have taken significant steps to address your suggestions in our manuscript. We would like to respond to your comments individually as following:

Complex Sentence Structure: we have restructured our sentences to enhance readability and ensure that readers can access essential information without relying extensively on cited references. Such as: (Line 16-18), (Line 305-307), and (Line 472-474).

Background Information: we have made significant revisions to provide more comprehensive background information, reducing the need for readers to refer extensively to cited references for context. Such as: (Line 20-23), (Line 43-48), and (Line 289-298).

bon Gene Cluster Prevalence: the prevalence of the bon gene cluster, affecting BKA production, has been highlighted to emphasize its significance in assessing food safety. Such as: (Line 20-23) and (Line 196-198).

Control Measures: we have included recommendations for control measures that align with the sporadic nature of BKA poisoning, particularly in traditional food settings with poor sanitation. Such as: (Line 263-270) and (Line 370-376).

In summary, your comments have significantly contributed to the refinement of our manuscript, and we are grateful for your expert guidance. We believe that these revisions have enhanced the overall quality and accessibility of our work.

“L.43: The word “respiratory toxin” should be explained in more detail.”

Response: thank you so much for the kind suggestion. The sentence has now been revised as “Bongkrekic acid (BKA) is a potent respiratory toxin, which could significantly impair mitochondrial ATP/ADP exchange and has been implicated in numerous outbreaks of fatal food poisoning.” to provide a more detailed explanation (Line 43-45).

“L.44-45: references should be cited for this statement”

Response: thank you. An appropriate reference has now been added (Line 45-46).

“L. 45-47: the reference cited did not support this sentence (bon “only existed in this bacterium”) to claim this statement, other bacteria should be assessed for this gene and found to be negative. maybe from other reference, please check”

Response: thank you. We apologize that the statement lacks certain level of rigors and has been revised and more comprehensive references have been added as “Previous research has demonstrated that BKA is produced by Burkholderia gladioli pv. cocovenenans in natural conditions seeing that the BKA biosynthesis gene cluster (bon) exists in the gene repertoire of this bacterium” (Line 46-49).

“Table 1: what is “noticeable” means in this context and why the main food, i.e., fermented coconut pulp was not in the list?”

Response: thank you. The use of “noticeable” could indeed rise confusions and the title of the table has been revised as “Documented bongkrekic acid poisoning cases.” (Line 69). While coconut-based products were predominantly associated with earlier cases of BKA poisoning, recent outbreaks have shifted towards corn- and rice-based foods. In this table, we have highlighted these outbreaks that have involved a higher number of cases and deaths to provide this pertinent information.

Figure 1: this figure should not be listed as a “framework”, it’s a list of findings. What is “certified literature” means in this context?

Response: thank you so much for the kind suggestions. The title of this figure has now been changed fully according to the suggestions as “Chronological list of significant research findings in BKA and B. gladioli pv. cocovenenans. Presented years are literature publication years.” (Line 143-144)

“Figure 3: Need more detail explanation. All arrows should be explained.”

Response: thank you. The arrows represent the conversion cycles between c-state and m-state of ANT thus allow the ADP/ATP exchange to be carried out. New explanations are now added to illustrate the function and impairment of conversion cycles: “When consumed by animals, owing to its apparent lipophilicity, BKA can firstly penetrate the inner mitochondrial membrane and then bind to m-state ANT to form BKA-inhibited ANT structure (Figure 3B)” (line 166-168).

“L. 204: This reference did not state this sentence.”

Response: thank you. This sentence has been rephrased as “This study revealed that out of 239 B. gladioli genomes, 36 contained the bon gene cluster.” for accuracy (Line 202-203).

“Figure 4: Please check for accuracy. The reference stated that no bon cluster in chromosome I in BSR3”

Response: thank you. Although the DNA preparation and high-throughput DNA sequencing were not carried out by the authors. Both circle genomes of B. gladioli BSR3 and Co14 were constructed according to genome assemble from NCBI database (can be found under the accession number of PRJNA64503 and PRJNA500835). And Figure 4 accurately represents respective gene allocations.

“L.246-247: Please check for accuracy”

Response: we genuinely appreciate the important observations made by the reviewer. The 3 open reading frames (ORFs)-bon gene cluster in B. gladioli pv. cocovenenans DSMZ 11318 (studies by Moebius and others [1]) was indeed a rather unique version of bon. Most other B. gladioli pv. cocovenenans strains contains a bon gene cluster with 4 ORFs with very similar overall identification.

“L. 286-288: stated that coconut oil and glycerol significantly impact the BKA production, however, Table 1 shows the occurrence of outbreaks caused by non-fatty food such as corn flour, rice noodles, and alcoholic drinks. This conflict should be discussed.”

Response: thank you so much for the insightful comment. The coconut oil and glycerol significantly impact (more than likely increase) the BKA production may explain the earlier BKA poisoning cases that occurred in coconut-related foods. Whereas modern BKA outbreaks (highly associated with corn- and rice-based foods) may be explained by factors such as pathogen transmission, outbreak countries (southeast Asia to China), prolonged fermentation period, supplied nurturant (sugar, protein, and/or fat) in respective foods. This conflict is an important observation and very much within the scope of future studies. A detailed discussion has now been added here (Line 289-298).

“Table 2 and its explanation in the main text is difficult to understand.”

Response: we attempted to comprehensively present the the relationship between substrate concentrations and substantial Bongkrekic acid (BKA) production in culture media. The difficulty arises from the fact that existing literature measures BKA production capacity from various angles, including inhibiting microbial growth, suppressing the bon gene cluster's BKA production, and/or promoting BKA production.

Rather than solely focusing on whether a specific substance at a certain concentration can produce BKA, we chose to explore the correlation between these substances and BKA production. It's important to note that the inhibition of BKA production doesn't necessarily align perfectly with whether B. gladioli pv. cocovenenans is still actively growing or surviving. Therefore, we used Table 2 as to consolidate this information. By documenting the literature's findings on the concentrations of certain substances, especially those harmful to the bacterial strains, at which BKA production is inhibited, we aimed to assess the independent inhibitory effects of such substances on the bon gene cluster activities. This approach goes beyond conventional antimicrobial activity, as it considers whether the microbial population continues to grow in the presence of these substances.

It's worth noting that the referenced literature dates back at least 30 years and new research in this area holds significant future implications and underscores the need for further investigation into these complex interactions. Respective paragraph has been updated according to the reviewer’s comment for improvement (Line 289-327).

References:

  1. Moebius, N.; Ross, C.; Scherlach, K.; Rohm, B.; Roth, M.; Hertweck, C. Biosynthesis of the respiratory toxin bongkrekic acid in the pathogenic bacterium Burkholderia gladioli. Chem. Biol. 2012, 19, 1164-1174.

Reviewer 3 Report

Comments and Suggestions for Authors

an interesting risk profile for a rare pathogen. It seems that the cases of food transmission are due to inadequate control of preparation hygiene, leading to quantitatively high levels of contamination and, consequently, to the accumulation of high doses of bongkrekic acid. Basic hygienic conditions in the production and preparation of traditional coconut- and corn-based dishes appear to be sufficient to control this risk. It's a bit of an exaggeration to speak of a global threat, and this aspect should be downplayed in the text.

Author Response

Response: Dear reviewer, we genuinely appreciate your thoughtful and insightful comment. Your observation regarding the cases of food transmission and the role of inadequate preparation hygiene leading to high levels of contamination and bongkrekic acid accumulation is in complete agreement with our opinion. Regarding the mention of a "global threat," we revised the text to provide a more balanced view to accurately reflects the scope the issue (Line 20-23, Line 472-485). Again, we would like to express our heartfelt gratitude.

Round 2

Reviewer 3 Report

Comments and Suggestions for Authors

Author's changes are satisfactory